# The long-solved problem of the best-fit straight line: Application to isotopic mixing lines

Richard Wehr[1], Scott R. Saleska[1]

[1]Department of Ecology and Evolutionary Biology, University of Arizona, Tucson, 85721, U.S.A.

*Correspondence to*: Richard Wehr (rawehr@email.arizona.com)

**Abstract.** It has been almost 50 years since York published an exact and general solution for the best-fit straight line to independent points with normally distributed errors in both $x$ and $y$. York's solution is highly cited in the geophysical literature but almost unknown outside of it, so that there has been no ebb in the tide of books and papers wrestling with the problem. Much of the post-1969 literature on straight-line fitting has sown confusion not merely by its content but by its very existence. The

optimal least-squares fit is already known; the problem is already solved. Here we introduce the non-specialist reader to York's solution, and demonstrate its application in the interesting case of the isotopic mixing line, an analytical tool widely used to determine the isotopic signature of trace gas sources for the study of biogeochemical cycles. The most commonly known linear regression methods—ordinary least squares regression (OLS), geometric mean regression (GMR), and orthogonal distance regression (ODR)—have each been recommended as the best method for fitting isotopic mixing lines. In fact, OLS, GMR, and

ODR are all special cases of York's solution that are valid only under particular measurement conditions, and those conditions do not hold in general for isotopic mixing lines. Using Monte Carlo simulations, we quantify the biases in OLS, GMR, and ODR under various conditions and show that York's general—and convenient—solution is always the least biased.

## 1 Introduction

A common analytical task in the physical sciences is to find the true straight-line relationship underlying independently

measured points with normally distributed measurement errors in both the ordinate $y$ and abscissa $x$. The literature on this topic is profuse, but much of it was outdated even before it was written. Looking through it, one discovers a menagerie of least-squares regression methods, none strictly appropriate to the task at hand. What one is unlikely to find, unfortunately, is the general, exact, and convenient least-squares solution published in 1969 by geophysicist Derek York (York, 1969).

York was motivated by rubidium-strontium isochrons but his landmark solution was universal. Unfortunately, while it became standard in geophysics, York's solution has remained largely unknown to the broader scientific community: York's paper has been cited almost 2000 times within the geophysical literature and only about two dozen times in all the rest of the scientific literature combined. Meanwhile, the number of papers and chapters expounding on the subject as if the solution did not exist is staggering. Examples include widely consulted books like *Biometry* (Sokal and Rohlf, 1995) and *Numerical Recipes* (Press et al.,

2007) as well as articles in fields as diverse as: anthropology (Smith, 2009), water resources (Hirsch and Gilroy, 1984), clinical chemistry (Stöckl et al., 1998), marine biology (Laws and Archie, 1981), aerosol science (Leng et al., 2007), and astronomy (Feigelson and Babu, 1992).

One scientific problem wanting for York's solution is the isotopic mixing line. In 1958, Charles Keeling (Keeling, 1958)

introduced the idea of using an isotopic mixing line to determine the stable isotopic signature of a trace gas source. If the isotopic

composition $\delta$ of a trace gas with a net source or sink is plotted versus the inverse of its mixing ratio in air $c$, the points describe a straight line whose $y$-intercept gives the desired source or sink signature. The "Keeling plot" became a key method in isotope ecology and biogeochemistry, and is regularly used, for example, to determine the isotopic composition of respired $CO_2$. Unaware of York's general least-squares solution, researchers concerned with Keeling plots have debated which of the more widely known, special-case least-squares regression methods is best. Some studies have concluded in favor of ordinary least squares regression (OLS) (Zobitz et al., 2006; Kayler et al., 2010), others in favor of orthogonal distance regression (ODR) (Ogée et al., 2003; Miller & Tans, 2003), and still others in favor of geometric mean regression (GMR) (Pataki et al., 2003; Miller & Tans, 2003; Kayler et al., 2010). The disagreement arises because in each of these special cases, the measurement conditions influence the fit line bias.

Here we introduce the reader to York's solution and its practical application, using the isotopic mixing line as a concrete example. We then use Monte Carlo simulations to precisely quantify the biases inherent to York's solution and to the popular special-case regression methods under various common (and some uncommon) conditions. In Appendix A, we provide a short, fast, and easily implemented algorithm for computing York's best-fit slope and intercept, as well as their associated uncertainties.

## 2 Background

### 2.1 The taxonomy of straight-line fitting methods

The goal of straight-line fitting is to retrieve the true straight-line relationship between two variables, $x$ and $y$, from their measured values, $\hat{x}$ and $\hat{y}$ (hats will denote measured values throughout this paper). Though it is common to neglect the error in $\hat{x}$, doing so can lead to a biased fit, as both $\hat{x}$ and $\hat{y}$ are measurements and therefore corrupted by error at some level. The nature of that error might vary from point to point (a situation known as heteroscedasticity), and it might also be that the error in $\hat{x}_i$ is correlated with that in $\hat{y}_i$, where the subscript $i$ specifies a measurement pair (i.e. a data point). Such correlation can arise, for example, if both $\hat{x}$ and $\hat{y}$ are derived from the same quantity or measured by the same apparatus, and can be described by a (Pearson's) correlation coefficient $r$ that might also vary from point to point. Finally, it is conceivable that the errors in the various $\hat{x}_i$ might be correlated with one another, and similarly for the $\hat{y}_i$; that is, the points might not be independent.

If the points are independent of one another and their errors are normally distributed, then the problem can be treated by Least-Squares Estimation (LSE), which is equivalent to Maximum Likelihood Estimation (MLE) (Myung, 2003) in this situation. Most of the literature on straight-line fitting concerns LSE, as it is appropriate to the vast majority of straight-line fitting problems.

York's solution is the general LSE solution: his equations provide the best possible, unbiased estimates of the true intercept $a$ and slope $b$ in all cases where the points are independent and the errors are normally distributed. We refer to this general solution as exact because no approximation is involved in its derivation, although it cannot be written analytically and must therefore be obtained in practice by numerical iteration. In contrast, the LSE methods commonly used and debated—namely OLS, ODR, and GMR—provide unbiased estimates only in very specific special cases in which the solution can be written analytically. York (1966, 2004) considers those situations mathematically and shows that his equations reduce algebraically to OLS, GMR, etc. when appropriate.

Before discussing OLS, ODR, and GMR further, we should note that each is known by other names that add confusion to the literature (York, 1966; Hirsch and Gilroy, 1984): OLS is called 'the regression of $y$ on $x$', ODR is called 'major axis regression', and GMR is called 'reduced major axis regression'; ODR has also been called 'least normal squares' and 'the line of closest fit', while GMR has also been called 'the line of organic correlation', 'the unique solution', and 'the equivalence line'. The methods are also often categorized, with methods that consider error only in $\hat{y}$ (e.g. OLS) being called 'Model I' regressions, and those that consider error in both $\hat{x}$ and $\hat{y}$ (e.g. ODR, GMR) being called 'Model II' regressions (Sokal and Rohlf, 1995).

OLS is by far the most widely known fitting method. The OLS fit line is unbiased only when there is negligible error in $\hat{x}$ and when the error variance for the $\hat{y}_i$ does not vary with $i$. In this case, the problem reduces to minimizing the sum of the squares of the vertical distances of the points from the fit line. A variant known as weighted least-squares allows the error variance for the $\hat{y}_i$ to vary with $i$.

The ODR fit line (Pearson, 1901) is unbiased only when the error variances for the $\hat{x}_i$ and the $\hat{y}_i$ are equal and independent of $i$, and when the error in $\hat{x}_i$ is uncorrelated with that in $\hat{y}_i$. (The technique Pearson invented to arrive at the ODR fit line is called principal components analysis, which is widely used in its own right.) In this case, the problem reduces to minimizing the sum of the squares of the perpendicular distances of the points from the fit line. However, the condition of equal error variances is almost never satisfied in reality. The excessively restrictive nature of ODR is highlighted by the fact that the ODR result is not invariant under a change of scale. In other words, if one scales the y-axis by a factor of 10, the fit line slope does not scale by a factor of 10. This flaw led to the development of GMR (Kermack and Haldane, 1950), which effectively performs ODR on transformed coordinates: the $\hat{x}_i$ divided by the standard deviation of the $\hat{x}_i$, and the $\hat{y}_i$ divided by the standard deviation of the $\hat{y}_i$ (not by the standard deviations of their errors). However, GMR is also highly restrictive, as its fit line is unbiased only in the peculiar circumstance that the variance of the $\hat{x}$ error divided by the variance of the $\hat{x}_i$ is equal to the variance of the $\hat{y}$ error divided by the variance of the $\hat{y}_i$.

Neither OLS, ODR, nor GMR uses estimates of the actual measurement uncertainty in its determination of the best-fit line.

A superior fitting method, called *fitexy*, is provided as an algorithm in Press et al. (2007) and its earlier editions. Press et al. (2007) started (unknowingly) from a similar point as did York (1969), seeking to minimize a $\chi^2$ function that is identical to $S$ in Eq. (2) of York (1969) if the correlation coefficient there is set to zero. However, rather than taking advantage of York's (1966, 1969) algebra, Press et al. (2007) treat the problem as one of nonlinear minimization. This approach leads them to a more complicated (and slower) algorithm than one based on York's solution (such as that provided here in Appendix A). Moreover, *fitexy* is less general than York's solution in that it does not allow the errors in $\hat{x}_i$ and $\hat{y}_i$ to be correlated. The *fitexy* method has sometimes been confused with ODR in the literature (Ogée et al., 2003; Pataki et al., 2003).

If the errors in any or all of the $\hat{x}_i$ and $\hat{y}_i$ are known to be distributed non-normally, then strictly speaking, LSE does not apply and one should retreat to the even more general formalism of MLE. However, the bias introduced into York's solution by non-normal error distributions is not always a problem, as we show in Section 4.1. The bias will vary on a case-by-case basis, but if the correct distribution is known, then the significance of the bias can be estimated fairly simply by Monte Carlo simulation, as we do here.

If LSE must be rejected, MLE may or may not be tractable. MLE requires that the correct error distributions be written analytically, and that a useable expression be derived for the likelihood function $L$ appropriate to those distributions. We say 'useable' because simply applying the definition of $L$ to the case of a straight line with errors in both $\hat{x}$ and $\hat{y}$ yields an expression that requires knowledge of the true values $x_i$ and is therefore useless in and of itself. Expressing the likelihood function in a form that does not include the true values $x_i$ was essentially York's first step in deriving his solution (see Eqs. (7) through (14) of York (1966), in which $S \propto -\ln(L)$).

## 2.2 A note on natural variability

York's method deals with the situation in which there is a linear relationship between the true values $x$ and $y$ but those values are measured with random error. When the scatter of the measured values $\hat{x}$ and $\hat{y}$ about a straight line exceeds the uncertainty attributed to the measurement technique, the excess scatter is attributed to "natural variability". Natural variability may sometimes be describable as just another part of the measurement error; that is, as a stochastic process that intervenes between the quantity of interest and the measurement of that quantity. For example, a technique called eddy covariance is commonly used to estimate the flux of $CO_2$ through the two-dimensional plane overlying an ecosystem based on a single-point measurement on a tower, and most of the noise in the estimation comes not from the instrumental measurement uncertainty but from the natural (turbulence-driven) variability in the flux past that single point relative to the flux through the whole plane (Wehr et al., 2013). If the natural variability in $\hat{x}$ and the natural variability in $\hat{y}$ are describable as normally distributed measurement error and can be characterized independently (along with any correlation between them), then York's method can be applied and is likely to be very useful.

On the other hand, it is often the case that the natural variability is not well characterized, or that it is not well described as additional measurement error. In this case, we argue that one cannot proceed to determine the best-fit line, or even to define what "best-fit" means. In general, one can view natural variability as variation in the true $x$-$y$ relationship due to the influence of other factors that are not controlled for. So a Keeling plot with natural variability is like many true mixing lines all superimposed on the same plot (one line for each set of influencing factors). It is therefore pertinent to consider which true line one is looking for. To define that line is, in effect, to characterize the natural variability in $x$ and $y$.

If one is interested not in the $x$-$y$ relationship per se (i.e. not in an intercept or slope), but simply in predicting the most likely value of $\hat{y}$ given $\hat{x}$ for the particular data that were sampled and put on the plot, then OLS is the proper fit to use. If differences among the various fit methods are not large enough to matter to the scientific question being posed, then OLS is again a sensible choice, owing to its simplicity.

## 2.3 The isotopic mixing line problem: Keeling and Miller/Tans plots

A Keeling plot is a scatterplot of the stable isotopic composition $\delta$ of a trace gas (typically $CO_2$) versus the inverse of its mixing ratio $c$ in air (Fig. 1). The isotopic composition $\delta$ is defined as a relative deviation from a reference material: $\delta = (R - R_{ref})/R_{ref}$, where $R$ is the ratio of the rare isotope (e.g. $^{13}C$) abundance to the most common isotope (e.g. $^{12}C$) abundance, and $R_{ref}$ is that ratio for the reference material. The standard units for $\delta$ are ‰ (parts per thousand), while those for $c$ are ppm (parts per million). It is easy to show that in the case of a source or sink changing the trace gas mixing ratio in the atmosphere, the $y$-intercept of a straight line fit to a Keeling plot gives the isotopic signature (i.e. the $\delta$) of that source or sink. To wit, if the subscripts $t$, 0, and $s$ refer to the atmosphere at some time $t$, the atmosphere at time 0, and the source, respectively, then:

$$c_t = c_0 + c_s$$

$$\delta_t c_t = \delta_0 c_0 + \delta_s c_s = \delta_0 c_0 + \delta_s (c_t - c_0)$$

$$\Rightarrow \delta_t = \delta_s + (\delta_0 c_0 - \delta_s c_0) \times (1/c_t)$$

Multiplying the Keeling fit line equation $\delta = a + b(1/c)$ through by $c$ gives an alternate straight-line equation, $\delta c = b + ac$, which can be fit to a plot of $\delta c$ versus $c$, called a Miller/Tans plot (Miller & Tans, 2003). In that case, it is the slope of the line that gives

the source or sink signature. (Ironically, Keeling reported the inverse relationship between $\delta$ and $c$ in 1958 but no "Keeling plots" appear in any of his early papers, nor does he discuss straight line fitting *per se*; he chose instead to show curved fit lines to plots of $\delta$ versus $c$.)

In many ecosystems, the source/sink signatures of interest differ from one another by just 1 ‰ or less (Bowling et al., 2014), so

that Keeling or Miller/Tans plot fit biases of 0.1 ‰ can be important.

The first studies to consider the effect of error in $c$ on isotopic mixing line fits (Miller & Tans, 2003; Pataki et al., 2003; Ogée et al., 2003) advocated use of ODR or GMR on the grounds that OLS is biased by neglecting error in $c$; however, they quantified only the differences between OLS and the other methods rather than their true biases. Zobitz et al. (2010) then examined the true

biases through Monte Carlo simulations and reported that OLS was negligibly biased for all measurement conditions, and that the difference between OLS and GMR was in fact due to bias in GMR. Kayler et al. (2010) later revisited the issue and reported that OLS could be non-negligibly biased for both low (< ~50 ppm) and high (> 1000 ppm) $CO_2$ ranges, depending on the measurement conditions, and they advocated use of GMR on a Miller/Tans plot for the most accurate fits when the $CO_2$ range is high. One goal of the present article is to inform readers that there is a single general fit method (York's) that is best in all

measurement scenarios, so that they do not need to make a choice among biased methods or to switch between those methods depending on the conditions. Another goal is to detail the conditions under which York's general solution can be satisfactorily approximated by OLS, as we recognize that OLS will likely remain more accessible and familiar to most researchers.

## 3 Methods

### 3.1 Applying York's solution to isotopic mixing lines

An algorithm that solves the York equations (see Appendix A) requires as input, for each data point $i$: the abscissa $\hat{x}_i$, the ordinate $\hat{y}_i$, the standard deviations of the errors for $\hat{x}_i$ and $\hat{y}_i$, and a correlation coefficient $r_i$ describing the correlation between the error in $\hat{x}_i$ and that in $\hat{y}_i$. This set of information is the same set required for any accurate straight-line fit—if a method does not ask for some of these parameters, then it is implicitly assuming values for them. However, as York's equations must be solved by numerical iteration, they require one additional piece of information: an initial guess slope. This guess can be very

rough and still sufficient, as convergence is not sensitive to the value chosen, and the solution can be iterated to an arbitrary degree of accuracy (using the previous best-fit slope as the new guess slope). In practice, ten iterations are almost always sufficient. If desired, a good initial guess slope can be obtained from an OLS fit.

The accuracy of the best-fit line will depend on the accuracy of the error and correlation estimates, but almost any reasonable

estimates will be better than the estimates implicit in OLS, GMR, and ODR. (And good error estimates are to be sought anyway,

as a measurement is only meaningful to the extent that its uncertainty has been quantified.) The Keeling plot is an interesting application partly because, if we take the errors in $\hat{c}$ to be normally distributed, then the errors in the Keeling abscissa $\hat{x} = 1/\hat{c}$ are not normally distributed—though the errors in $\hat{c}$ tend to be small relative to $\hat{c}$ itself, so that the resulting bias in the fit line tends to be negligible (see Section 4.1). Miller/Tans plots are also interesting, for a different reason: though we take the errors in

$\hat{\delta}_i$ and $\hat{c}_i$ to be uncorrelated (which is usually the case in practice), the errors in the Miller/Tans abscissa $\hat{x}_i = \hat{c}_i$ and ordinate $\hat{y}_i = \widehat{(\delta c)}_i = \hat{\delta}_i \hat{c}_i$ are always correlated because both contain $\hat{c}_i$. Moreover, the errors in the Keeling abscissa and Miller/Tans ordinate are both (slightly) heteroscedastic.

Regarding correlation between the errors in $\hat{\delta}_i$ and $\hat{c}_i$, it is perhaps worth mentioning that the high-frequency (e.g. 1 Hz) random

noise in modern spectroscopic measurements of $\delta$ and $c$ is generally correlated due to the fact that certain causes of random spectroscopic noise (e.g. laser frequency instability) necessarily affect retrievals of both $\delta$ and $c$. The correlation may nonetheless be negligible for data averaged over a minute or longer.

The standard deviations of the errors in $\widehat{\delta c}$ and $1/\hat{c}$, as well as the Miller/Tans correlation coefficient, are given by Eqs. (1), (2),

and (6), below. For a series of $i$ measurement pairs $\{\hat{c}_i, \hat{\delta}_i\}$, let the standard deviations of the normally distributed measurement errors be $\{\varepsilon_i, \eta_i\}$. Then the standard deviation of the measurement errors in the Miller/Tans ordinate $\widehat{(\delta c)}_i$ is given by (again, assuming no correlation between the errors in $\hat{\delta}_i$ and $\hat{c}_i$):

$$\phi_i = \sqrt{\varepsilon_i^2 \delta_i^2 + \eta_i^2 c_i^2} \approx \sqrt{\varepsilon_i^2 \hat{\delta}_i^2 + \eta_i^2 \hat{c}_i^2}. \tag{1}$$

Meanwhile the standard deviation of the (not really normally distributed) measurement errors in the Keeling abscissa $(1/\hat{c})_i$ is given by:

$$\theta_i = \varepsilon_i / c_i^2 \approx \varepsilon_i / \hat{c}_i^2. \tag{2}$$

Eqs. (1) and (2) are standard expressions for the propagation of uncertainty through the operations of multiplication and division. For a Keeling fit, the correlation coefficient $r_i$ should be set to zero for all $i$ (unless there is correlation between the measurement errors in $\delta_i$ and $c_i$). For a Miller/Tans fit, $r_i$ is defined as:

$r_i \stackrel{\text{def}}{=} \text{cov}((\delta c)_i', c_i') / \phi_i \varepsilon_i = \langle (\delta c)_i' c_i' \rangle / \phi_i \varepsilon_i. \tag{3}$

In Eq. (3), $\text{cov}(A, B)$ denotes the covariance of $A$ and $B$, the angle brackets denote expectation value, and we have decomposed the measured values as $\hat{c}_i = c_i + c_i'$, $\hat{\delta}_i = \delta_i + \delta_i'$, and $\widehat{(\delta c)}_i = (\delta c)_i + (\delta c)_i'$, where primes denote measurement errors and unaccented variables denote true values. Note that the expectation value here is not an average over the data points, which would

involve variations in the true $\delta$ and $c$. It is rather the expectation value for a given data point, i.e. what the value for the data point would be if it were an average of many measurements with the true $\delta$ and $c$ held constant.

It follows immediately from the above decompositions that:

$$(\delta c)'_i = \delta_i' c_i + c_i' \delta_i + c_i' \delta_i',$$

(4)

and so:

$$\langle (\delta c)_i' c_i' \rangle = \langle c_i' \delta_i' c_i + (c_i')^2 \delta_i + (c_i')^2 \delta_i' \rangle$$
$$= \delta_i \langle (c_i')^2 \rangle$$
$$= \delta_i \varepsilon_i^2$$
$$\approx \hat{\delta}_i \varepsilon_i^2$$

(5)

where the second line follows because none of the variables is correlated with another. Thus the correlation coefficient is given by:

$$r_i = \hat{\delta}_i \varepsilon_i / \phi_i.$$

(6)

So for the Keeling plot, we have $\hat{x}_i = 1/\hat{c}_i$ with uncertainty $\theta_i$, $\hat{y}_i = \hat{\delta}_i$ with uncertainty $\eta_i$, and $r_i = 0$. For the Miller/Tans plot, we have $\hat{x}_i = \hat{c}_i$ with uncertainty $\varepsilon_i$, $\hat{y}_i = \hat{\delta}_i \hat{c}_i$ with uncertainty $\phi_i$, and $r_i = \hat{\delta}_i \varepsilon_i / \phi_i$.

The approximations in Eqs. (1), (2), and (5) are usually excellent. Despite the precision afforded our Monte Carlo simulations by using $2.5 \times 10^7$ data points, we detect bias due to these approximations only under extreme circumstances (see Section 4.1).

On a modest 2009-model notebook computer, using the Igor Pro programming language (WaveMetrics, Inc.), 5000 five-iteration York fits to a 5000-point mixing line took 215 seconds (compare OLS at 14 seconds and *fitexy* at 1410 seconds), while 100,000
such fits to a 20-point mixing line took just 25 seconds (compare OLS at 66 seconds and *fitexy* at 150 seconds). For small numbers of fits, all of these methods are effectively instantaneous. It is perhaps ironic that for most real-world Keeling or Miller/Tans plots, involving fewer than 100 points, solving the York equations is actually faster than OLS.

**3.2 Monte Carlo simulations**

We tested the York, OLS, ODR, GMR, and *fitexy* methods using simulated measurements of an isotopic mixing line typical of
$CO_2$ respiration in a forest. In this way, the true line is known and the fit bias can be assessed. Our true mixing line had a source isotopic signature of exactly -25 ‰, a background $c$ of 380 ppm, and a background $\delta$ of -9 ‰. (A Keeling plot of this line has a slope of 6080 and a *y*-intercept of -25.) We simulated measurements of this line for a variety of mixing ratio ranges and measurement precisions. For each set of conditions, we generated 5000 'measured' lines, each comprising 5000 points spread evenly over the mixing ratio range $\Delta c$, with normally distributed, uncorrelated errors added to $c$ (with standard deviation $\varepsilon$) and
to $\delta$ (with standard deviation $\eta$). The same values of $\varepsilon$ and $\eta$ were used for all points. We then expressed each line as a Keeling plot and as a Miller/Tans plot, and fit each plot by each of the methods.

Real-world mixing line plots are not likely to contain 5000 points each, but using a large number of points per plot can be important when precisely quantifying fit bias in an ensemble of lines. A demonstration of this point is provided in Appendix B

for the interested reader. Uninterested readers will be content to know that using more points per plot does not add any new bias to the results—although it might clarify existing bias from an inappropriate fit model, which is what is going on in the discussion under "Sample size effect" in Kayler et al. (2010).

We performed tests for $\Delta c$ = 1, 5, 10, and 50 ppm, and for various combinations of $\varepsilon$ (ppm) and $\eta$ (‰): $\varepsilon$ = 0.01, $\eta$ = 0.01 (slightly better than the best existing laser spectrometer; Wehr et al., 2013); $\varepsilon$ = 0.2, $\eta$ = 0.3 (for a popular commercial laser spectrometer); $\varepsilon$ = 0.05, $\eta$ = 0.05 (for some pairings of an infrared gas analyzer (IRGA) and an isotope ratio mass spectrometer (IRMS)); and $\varepsilon$ = 0.15, $\eta$ = 0.01 (for other IRGA/IRMS pairings). Because Kayler et al. (2010) were concerned with biases at very large $\Delta c$ and $\varepsilon$ in studies of leaf respiration and the like, we also conducted simulations for $\Delta c \geq$ 100 ppm with $\eta$ = 0.2 ‰ as
in Kayler et al. (2010) and with $\varepsilon$ = 1, 5, and 20 ppm.

### 3.3. Forest air measurements

In addition to our Monte Carlo simulations, we analyzed 429 Keeling plots composed of real measurements, specifically nighttime measurements of the mixing ratio and $^{13}C$ composition of $CO_2$ in the air within and above a forest canopy. The intercept of such a Keeling plot should give the isotopic composition of nighttime respiration. The measurements were made at
each of 6 or 7 heights on a 29 m tower every 40 or 45 minutes from May through October of 2011, 2012, and 2013, as described elsewhere (Wehr et al., 2013; Wehr & Saleska, 2015). Each Keeling plot was made from one night's data and included about 50 points. The measurement precisions were $\varepsilon$ = 0.05 ppm and $\eta$ = 0.02 ‰. We also did an alternate analysis in which additional random noise of 0.2 ppm and $\eta$ = 0.3 ‰ (corresponding to more common spectroscopic instrumentation) was added to the measurements.

## 4 Results and discussion

### 4.1 Comparison of fit biases using Monte Carlo simulations

Isotopic source signatures retrieved from our simulated Keeling plots for $\Delta c \leq$ 50 ppm are reported in Table 1. The main cells contain three numbers: the upper is the York result (in bold), while the middle is the OLS result, and the lower is the GMR result. We have not tabulated the *fitexy* or ODR results; the *fitexy* results are discussed below, and it is enough to say that the
ODR results were no better than the GMR results (ODR being a seriously flawed precursor to GMR, as explained in Section 2.1). The units of ‰ here (and throughout this paper) are simply the units of $\delta$, and not an indication of relative error in the results. Numbers in parentheses represent the standard error in the last digit, calculated from the distribution of retrieved values rather than by York's equations, although the two results did agree closely (see Section 4.3). We have omitted our Miller/Tans results from Table 1 because there were no significant differences between the Miller/Tans and Keeling results for the York and
OLS methods, and because the GMR results were very poor for both plot types.

Indeed, as reported by Zobitz et al. (2006), GMR produces highly biased fit lines (that is, the retrieved source signature falls much more than 3 standard errors from -25 ‰). For $CO_2$ ranges less than 50 ppm, the GMR bias is non-negligible in practical terms (> 0.1 ‰) unless the measurement uncertainty in $\delta$ is extraordinarily low ($\leq$ 0.01 ‰). OLS does better than GMR in most
of the tested circumstances because the relative error in the $c$ measurement is usually much less than the relative error in the $\delta$ measurement, but OLS still involves important levels of bias when $\varepsilon$ is large or $\Delta c$ is small.

The York equations, on the other hand, produce unbiased Keeling and Miller/Tans fit lines for all conditions in the table. Because the emergence of high-frequency isotopic measurements is starting to raise the issue, we show in detail how some OLS- and York-retrieved isotopic source signatures compare at the lowest $\Delta c$ in Figure 2, where the error bars represent $\pm 2\sigma$, i.e. twice the standard error in the mean of 5000 fits.

Isotopic source signatures retrieved from our simulated Keeling and Miller/Tans plots for $\Delta c \geq 100$ ppm are reported in Table 2. Again, the York method is by far the best, although the York fit lines do exhibit small but detectable biases for some sets of conditions here. The York Miller/Tans fits are biased by at most -0.020 ‰, while bias in the York Keeling fits is worse, reaching -0.204 ‰ when $\varepsilon = 20$ ppm (an exceptionally high value) and $\Delta c = 100$ ppm. This bias is still an order of magnitude less than the bias from any of the other methods under those conditions. The York Miller/Tans bias is due to the approximations made in Eqs. (1) and (5). We have confirmed this by comparing simulations with and without the approximations (a luxury not available with real data). The Keeling bias is due partly to the approximation in Eq. (2), but mostly to the non-normal error distribution in $1/\hat{c}$ (see Section 3.1): a distribution that is asymmetric, with a non-zero mean. In Table 1, where $\varepsilon$ (maximum 0.15 ppm) is always a small fraction of $c$ (380 ppm), the skew of the error in $1/\hat{c}$ is small and its effect on the fit negligible; however, in Table 2, where $\varepsilon/c$ can be as large as 5 %, the skew becomes relatively large (the uncertainty on one side of a point is about 10 % larger than on the other side) and its effect on the fit becomes detectible in our simulations. The bias induced in the fit by the non-normal error distribution should increase as $\varepsilon$ increases and as $\Delta c$ decreases, which are the trends we observe. The preceding explanation is confirmed by the fact that when we alter our simulations by adding normally distributed measurement error directly to $1/c$ rather than to $c$ (and giving the correct information to the York fitting algorithm), we find that the York Keeling fits are completely unbiased (results not tabulated). Luckily, the York fit biases we report in Table 2 are very small considering the measurement uncertainties necessary to induce them, and are unlikely to be the limiting source of error in any experiment.

We also tested *fitexy* using our Monte Carlo simulations. As expected, the *fitexy* results were always identical to the York results when fitting to Keeling plots but were in error when fitting to Miller/Tans plots, because the latter plots involved correlation between the $x$ and $y$ errors. For example, with the fairly large measurement uncertainties $\varepsilon = 0.2$ ppm and $\eta = 0.3$ ‰, the *fitexy* Miller/Tans slopes were biased by -4.259 ‰ for $\Delta c = 1$ ppm and -0.027 ‰ for $\Delta c = 10$ ppm.

### 4.2 Comparison of fit biases using real measurements

The intercepts from OLS, GMR, and York fits to our measured Keeling plots are compared in Figure 3, as a function of the $CO_2$ range. Given that our Monte Carlo simulations show the York fit to be unbiased, we can use the difference between the OLS (or GMR) and York intercepts as a proxy for the bias in OLS (or GMR). In agreement with our Monte Carlo results, Figure 3 shows that for the original measurement uncertainties of 0.05 ppm and 0.02 ‰, GMR is negligibly biased (that is, by less than 0.1 ‰) only for $CO_2$ ranges above about 25 ppm, while OLS is negligibly biased for all $CO_2$ ranges. Also in agreement with our Monte Carlo results, the figure shows that if the measurement uncertainties are increased to the more common values of 0.2 ppm and 0.3 ‰, then GMR is never negligibly biased, while OLS is negligibly biased only for $CO_2$ ranges above 10 ppm. Note that the scatter in Figure 3 is due to the fact that unlike our simulated data points, our real measured data points were not evenly distributed throughout the $CO_2$ range; in some of the measured Keeling plots, almost all of the points were clustered in a small portion of the range, leading to a higher bias.

### 4.3 Estimating the fit errors and the goodness of fit

York et al. (2004) provide compact equations not only for the slope and intercept, but for their standard errors as well. They further show that these error estimates are algebraically identical to those of the more general error formulation of MLE. Note, however, that while the York equations for the slope and intercept are exact (if the measurement uncertainties are normally distributed), the York/MLE estimates of the errors in the slope and intercept are approximations (Titterington and Halliday, 1973). In Table 3, we compare York's error estimates against the standard deviations of the Keeling plot intercepts retrieved from our Monte Carlo simulations. York's error estimates agree extremely well with the Monte Carlo results except when the measurement error variances are so large as to approach the total variances in $x$ and $y$ (i.e. when $\varepsilon/\Delta c$ and $\eta/\Delta\delta$ approach 1). Under those conditions, the agreement is nonetheless within 33%.

The errors estimated by *fitexy* (not tabulated) were slightly higher than those estimated by the York equations, tending to be farther from the Monte Carlo results for small $\varepsilon/\Delta c$ (when the line is well-constrained) and closer to the Monte Carlo results for large $\varepsilon/\Delta c$ (likely by coincidence). The York and *fitexy* estimated errors were of the same order of magnitude, however, and their disagreement may relate to a subtle point raised in York et al. (2004) concerning whether the errors are estimated using the original data points or what York calls the 'adjusted' data points, which are the fit method's reconstruction of the true, error-free data points.

The standard error is a measure of precision; it does not speak to how well the straight-line model represents the data—a concept known as goodness of fit. York et al. (2004) note that the goodness of fit of the York solution is estimated by the quantity $S/(R-2)$, where $R$ is the number of points in the fit and $S$ is given by:

$$S = \sum W_i(Y_i - bX_i - a)^2. \tag{7}$$

This goodness of fit metric is a weighted reduced chi-squared statistic, which we denote here by $\chi_W^2$, i.e. $\chi_W^2 = S/(R-2)$. $\chi_W^2$ is essentially comparing the deviations of the points from the fit line to the assigned measurement error standard deviations. If the variables $x$ and $y$ are in fact related by a straight line, and if the assigned measurement errors are correct (and normally distributed), then $\chi_W^2$ will equal 1. A value of $\chi_W^2$ significantly different from 1 indicates the failure of one or both of those assumptions, where we suggest that significance be defined relative to the standard error in $\chi_W^2$, which depends only on the number of points per fit, $R$, and is given by:

$$\sigma_\chi = \sqrt{2/(R-2)}. \tag{8}$$

We have tested this equation by Monte Carlo simulation and found it to be correct to within the simulation precision of roughly 1 part in 200.

The right-hand column of Table 3 gives the mean value of $\chi_W^2$ for each 5000-line ensemble described in the table. With $R = 5000$, our $\sigma_\chi$ equals 0.02, giving a standard error in our mean $\chi_W^2$ of $3\times10^{-4}$. Even at this level of precision, $\chi_W^2$ indicates that our fits are almost always good, as expected given the nature of our simulations. However, under those conditions for which the non-normal distribution of $1/c$ is a source of detectible bias, i.e. when $\varepsilon = 20$ ppm, we see the expected deviation in $\chi_W^2$, which

reaches a low of 0.986. In practice, without many fit lines of many points each, such a small drop in $\chi_W^2$ would be undetectable, as would the associated slope and intercept biases.

**4.4 Another application of York's solution: comparing two instruments**

When comparing measurements of the same quantity by two different instruments, it is common to plot the values obtained by one instrument against those obtained by the other, so that the relative bias between the instruments can be determined from a straight line fit to the plot. Monte Carlo simulations similar to those used for our isotopic mixing line example confirm that OLS and GMR may incorrectly estimate that bias. For example, if an old, unbiased instrument is being replaced by a new, also-

10 unbiased instrument whose precision is 5 times better, and if the two instruments are compared over a trial period in which the measured quantity varies over a range that is 20 times greater than the precision of the old instrument, then OLS (GMR) will incorrectly indicate that the new instrument is biased low by 4% (2%) of the reading. The York equations will correctly indicate no relative bias.

**5 Conclusion**

We have shown that the general least-squares solution for the best-fit straight line, published by Derek York in 1969, is the least biased estimator of the isotopic signature of a trace gas source from a Keeling or a Miller/Tans plot, regardless of the measurement conditions. In contrast, the popular regression methods considered in the literature are unbiased only under particular, often unrealistic conditions. The isotopic mixing line illustrates the virtue and convenience of York's solution, which is applicable to line fitting problems in many scientific disciplines. We have provided a short, fast pseudo-code algorithm for

computing York's solution, and derived simple equations for the required measurement error and correlation parameters in the case of a Keeling or Miller/Tans plot. Being both accurate and convenient, York's solution supersedes all other least-squares straight-line fit methods.

**Appendix A: Computer pseudo-code algorithm**

Here we provide an algorithm in pseudo-code for computing the slope and intercept of the best-fit straight line according to Eqs.

(13) of York et al. (2004). The data consist of $R$ data points indexed by the subscript $i$. The inputs are: the abscissa $X_i = \hat{x}_i$; the ordinates $Y_i = \hat{y}_i$; their measurement error standard deviations $\sigma_{X,i}$ and $\sigma_{Y,i}$; the correlation coefficient between those errors, $r_i$; and an initial guess slope $b_0$. The outputs are: the intercept $a$ and slope $b$, their standard errors $\sigma_a$ and $\sigma_b$, the goodness of fit $\chi_W^2$, and its standard error $\sigma_\chi$. The algorithm iterates until the slope converges to within the tolerance $T$. The fist block computes the slope and intercept, after which the second block computes the fit errors and the goodness of fit.

$b = b_0$

While $b_{\mathrm{diff}} > T$ do

Begin loop

$\quad b_{\mathrm{old}} = b$

$\quad \bar{X} = 0 \ \& \ \bar{Y} = 0 \ \& \ W_{\mathrm{sum}} = 0$

$\quad$ For $i = 1, R$ step 1 do

Begin loop

$$\omega_{X,i} = 1/\sigma_{X,i}^2 \ \& \ \omega_{Y,i} = 1/\sigma_{Y,i}^2$$

$$\alpha_i = \sqrt{\omega_{X,i}\omega_{Y,i}}$$

$$W_i = \alpha_i^2/(b^2\omega_{Y,i}) + \omega_{X,i} - 2br_i\alpha_i$$

$$\bar{X} = \bar{X} + W_iX_i \ \& \ \bar{Y} = \bar{Y} + W_iY_i$$

$$W_{\text{sum}} = W_{\text{sum}} + W_i$$

End loop

$$\bar{X} = \bar{X}/W_{\text{sum}} \ \& \ \bar{Y} = \bar{Y}/W_{\text{sum}}$$

$$Q_1 = 0 \ \& \ Q_2 = 0$$

For $i = 1, R$ step 1 do

Begin loop

$$U_i = X_i - \bar{X} \ \& \ V_i = Y_i - \bar{Y}$$

$$\beta_i = W_i\big[(U_i/\omega_{Y,i}) + (bV_i/\omega_{X,i}) - (bU_i + V_i)\,r_i/\alpha_i\big]$$

$$Q_1 = Q_1 + W_i\beta_iV_i \ \& \ Q_2 = Q_2 + W_i\beta_iU_i$$

5     End loop

$$b = Q_1/Q_2$$

$$b_{\text{diff}} = |b - b_{\text{old}}|$$

End loop

$$a = \bar{Y} - b\bar{X}$$

$$\bar{x} = 0$$

For $i = 1, R$ step 1 do

Begin loop

$$x_i = \bar{X} + \beta_i$$

$$\bar{x} = \bar{x} + W_ix_i$$

10     End loop

$$\bar{x} = \bar{x}/W_{\text{sum}}$$

$$\sigma_b = 0 \ \& \ \chi_W^2 = 0$$

For $i = 1, R$ step 1 do

Begin loop

$$u_i = x_i - \bar{x}$$

$$\sigma_b = \sigma_b + W_iu_i^2$$

$$\chi_W^2 = \chi_W^2 + W_i(Y_i - bX_i - a)^2$$

End loop

$$\sigma_b = \sqrt{1/\sigma_b}$$

$$\sigma_a = \sqrt{\bar{x}^2\sigma_b^2 + 1/W_{\text{sum}}}$$

$$\chi_W^2 = \chi_W^2/(R-2)$$

$$\sigma_\chi = \sqrt{2/(R-2)}$$

## Appendix B: Biased estimators and the sample size effect

Why simulate 5000 points per plot when real plots are more likely to contain 20 or 50 points? Consider the simple picture in Figure 4. Here a true line with slope 1 and $y$-intercept 0 passes through the points (2,2) and (6,6). Measurements at those two points contain errors in $y$ only, and the errors are supposed to be either +1 or -1. The two measured lines shown have errors that are symmetric in the sense that they are equal in magnitude and opposite in sign, and yet these measured lines are not symmetric about the true line in the sense of being reflections of one another in the true line. Such symmetry would arise only if there were equal error in both $x$ and $y$. Nonetheless, the measured slopes (1/2 and 3/2) and $y$-intercepts (2 and -2) are indeed symmetric about the true values. Now imagine reflecting this picture about $y = x$ (or rotate the page 90˚), so that the line is measured with errors of ±1 in $x$ only. In this case, the measured slopes (2/3 and 2) and $y$-intercepts (4/3 and -4) are highly asymmetric about the true values although the fit lines are just as good as before. Instead, the $x$-intercepts and inverse slopes have become the symmetric parameters. All this follows from the asymmetries inherent to the definitions of the slope and intercept.

Translating the simple picture of Figure 4 into practical reality, one finds that when there is normally distributed measurement error only in $y$, straight line fits can be characterized by their slopes and $y$-intercepts, which are normally distributed parameters that can be averaged over an ensemble of fits to give an unbiased estimate of the slope and $y$-intercept of the mean fit line (by which we mean the line that would be obtained by combining all of the measured data from the whole ensemble on one plot and doing a single fit to that). However, when error is present also (or only) in $x$, the slopes and $y$-intercepts of an ensemble of fit lines are not normally distributed about the mean-line values. Instead the distributions are skewed, with means and modes that differ from one another and from the mean-line values. In the general case, with uncertainty in both $x$ and $y$ that might vary from point to point, there are no standard fit line parameters that can be averaged to give the corresponding parameters of the mean fit line.

Fortunately, as the $CO_2$ range or the number of points per fit line becomes large, the deviations of the fit lines from the true line become small and the skew in the parameter distributions becomes insignificant. We exploited this fact in our Monte Carlo simulations by using 5000 points per line, which enabled us to simply average our fit slopes and intercepts without introducing significant bias even for $\Delta c = 1$ ppm. Real-world Keeling plots often contain only 10 or 20 points, but usually span more than 10 ppm, and so the skew remains small. Moreover, real-world Keeling plots are not found in ensembles of 5,000, and so the statistical error in the mean slope and intercept would swamp any skew-related bias even for $\Delta c = 1$ ppm.

One might be worried about increasing the number of points per plot because it has been reported that fit bias appears to increase with the number of points per plot (Kayler et al., 2010). Actually, increasing the number of points per plot merely clarifies the bias associated with a poor fitting method. Consider Figure 5, where we plot distributions of $y$-intercepts retrieved from Keeling plots with $\Delta c = 1$, $\varepsilon = 0.15$, and $\eta = 0.01$. These conditions are extreme in that $\varepsilon/\Delta c \approx \eta/\Delta\delta \approx 0.2$ and so the measurement error spread in each dimension is roughly equal to the extent of the true line. Effectively similar conditions ($\Delta c = 40$, $\varepsilon = 5$) were used to produce Table 3 of Kayler et al. (2010). In our figure, the thick curve is the distribution with just 2 points per line, in which case there is no distinction between fit methods because "fitting" consists merely of drawing a line exactly between the two measured points. As the number of points per line increases, they form a distribution in Cartesian space that each fit method interprets differently (according to its assumptions about the measurement error distributions and correlation), so that the intercept distributions obtained from different fit methods diverge. The thin solid curves in Figure 5 show the distributions retrieved from the York equations as the number of points per plot increases to 5 and then 20. The dashed curves show the

corresponding distributions retrieved from OLS. By 20 points, the distributions are nearly Gaussian, and the York distribution is nearly centered on the true mean while the OLS distribution is nearly centered on the value corresponding to its true fit bias under these uncertainty conditions. Note that the skew in Figure 5 is not due to the subtly skewed noise distribution in the abscissa: the figure is not visibly changed when normally distributed noise is added directly to $1/c$ rather than to $c$.

## Author contributions

R. Wehr designed and performed the study, and prepared the manuscript with contributions from S. R. Saleska.

## Competing interests

The authors declare that they have no conflict of interest.

## Acknowledgments

This research was supported by the US Department Of Energy, Office of Science, Terrestrial Ecosystem Science program (award DE-SC0006741), by the National Science Foundation Macrosystems Biology program (award #1241962), and by the Agnese Nelms Haury Program in Environment and Social Justice at the University of Arizona.

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

**Figures, Tables, and Captions**

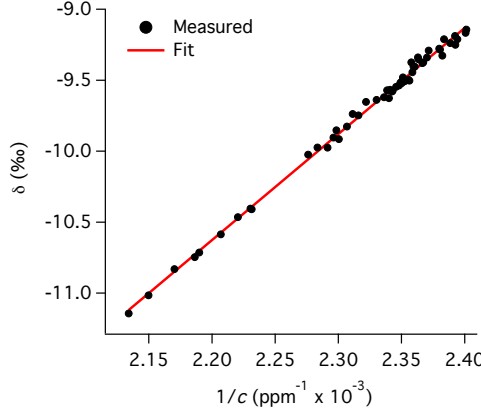

**Figure 1: An example Keeling plot from the set described in Section 3.3, showing 55 measurements made on the night of May 25, 2011,**
15    **spanning a $CO_2$ range $\Delta c = 52$ ppm, with measurement error standard deviations of $\varepsilon = 0.05$ ppm and $\eta = 0.02$ ‰.**

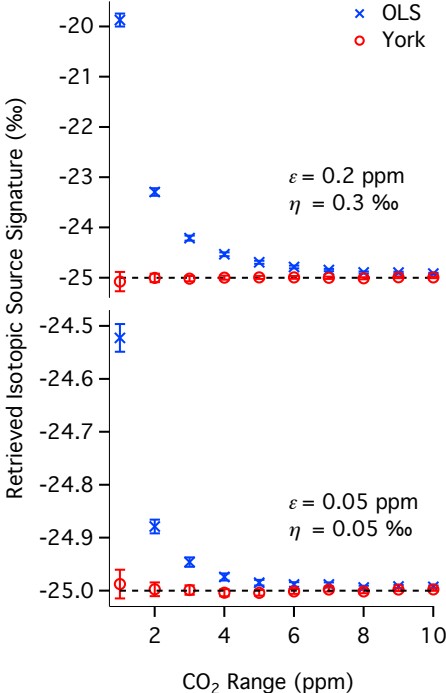

**Figure 2: Mean isotopic signatures retrieved from ensembles of 5000 simulated Keeling plots (each containing 5000 points) using the York and OLS methods, for CO$_2$ ranges from 1 to 10 ppm and a true isotopic signature of -25 ‰. The standard deviations of the random noise added to the simulated measurements of $c$ and $\delta$ are inset. Error bars show two standard errors. Note the differing scales for the ordinate.**

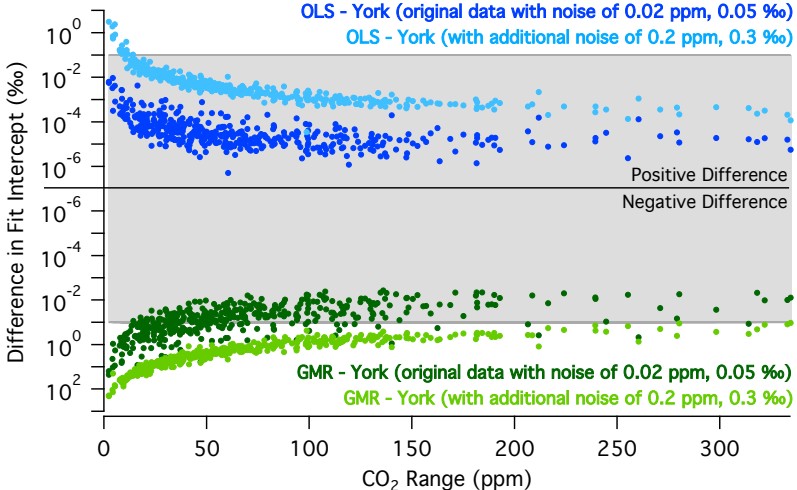

**Figure 3: Difference between the fit intercept obtained by OLS or GMR and that obtained by York's equations, for 429 real measured Keeling plots with measurement uncertainties of 0.05 ppm and 0.02 ‰ and with CO$_2$ mixing ratio ranges between 2 and 335 ppm. Also shown is the difference obtained for Keeling plots in which random computer-generated noise of 0.2 ppm and 0.3 ‰ was added to the original data. The grey region represents biases that are negligible for most practical purposes (< 0.1 ‰).**

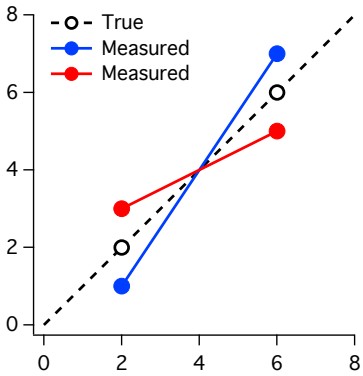

**Figure 4: Two lines, each defined by measurements of the true points (2,2) and (6,6) that are in error by +1 or -1 in the ordinate only.**

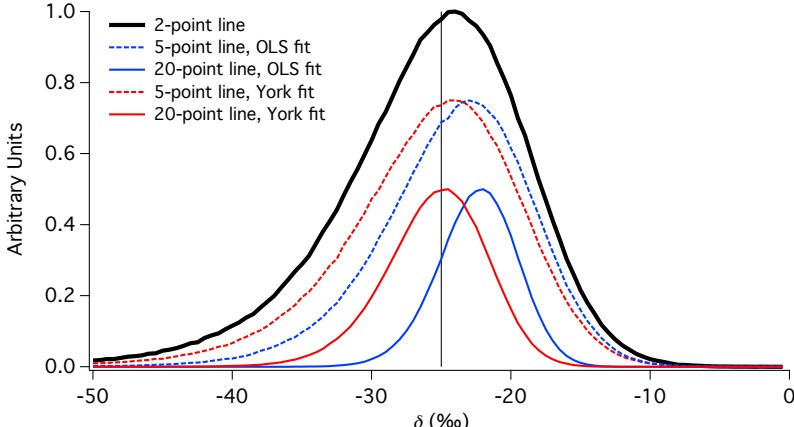

5    **Figure 5: Distributions of the *y*-intercepts retrieved from Keeling plots with a true intercept of -25 ‰ (indicated by the vertical line). The black curve was obtained from $10^6$ two-point plots. The red curves were obtained using the York equations on $10^6$ five-point or twenty-point plots. The blue curves were obtained using OLS on $10^6$ five-point or twenty-point plots. The heights of the curves have been adjusted independently for presentation and convey no meaning.**

**Table 1. Bias in retrieved isotopic source signatures from Keeling line fits for $\Delta c \leq 50$ ppm, expressed as differences from the true value of -25, in units of ‰. Each cell contains, from top to bottom: the York, the OLS, and the GMR result. 1-σ uncertainties in the final digit(s) are given in parentheses, based on the standard deviation of the ensemble of 5000 fits.**

| $\varepsilon$ (ppm) | $\eta$ (‰) | CO$_2$ Range (ppm) | | | |
| --- | --- | --- | --- | --- | --- |
| | | 1 | 5 | 10 | 50 |
| 0.01 | 0.01 | **0.005(3)** | **0.000(1)** | **0.000(1)** | **0.000(0)** |
| | | 0.024(3) | 0.001(1) | 0.000(1) | 0.000(0) |
| | | -4.700(3) | -0.214(1) | -0.054(1) | 0.002(0) |
| 0.01 | 0.15 | **-0.066(40)** | **-0.006(8)** | **0.007(4)** | **0.000(1)** |
| | | -0.047(40) | -0.005(8) | 0.008(4) | 0.000(1) |
| | | -181.95(3) | -26.592(6) | -9.400(4) | -0.477(1) |
| 0.05 | 0.05 | **0.010(14)** | **0.001(3)** | **0.004(2)** | **0.000(1)** |
| | | 0.475(13) | 0.020(3) | 0.009(2) | 0.000(1) |
| | | -50.840(10) | -4.781(3) | -1.343(2) | -0.065(1) |
| 0.15 | 0.01 | **0.002(4)** | **0.000(1)** | **0.000(1)** | **0.000(0)** |
| | | 3.398(3) | 0.171(1) | 0.043(1) | 0.002(0) |
| | | -2.385(3) | -0.128(1) | -0.032(1) | -0.001(0) |
| 0.15 | 0.15 | **-0.072(44)** | **-0.005(9)** | **0.001(4)** | **0.000(1)** |
| | | 3.339(35) | 0.165(8) | 0.043(4) | 0.002(1) |
| | | -160.00(3) | -26.673(6) | -9.657(4) | -0.575(1) |
| 0.2 | 0.3 | **0.108(97)** | **-0.028(16)** | **-0.006(8)** | **0.002(2)** |
| | | 5.251(65) | 0.272(16) | 0.070(8) | 0.005(2) |
| | | -303.09(89) | -64.388(12) | -26.100(6) | -2.185(2) |

**Table 2. Bias in retrieved isotopic source signatures from Keeling and Miller/Tans line fits for $\Delta c \geq 100$ ppm and $\eta = 0.2$ ‰, expressed as differences from the true value of -25, in units of ‰. Each cell contains, from top to bottom: the York, the OLS, and the GMR result. 1-$\sigma$ uncertainties in the final digit are given in parentheses, based on the standard deviation of the ensemble of 5000 fits.**

| | CO$_2$ Range (ppm) | | | | | |
|---|---|---|---|---|---|---|
| $\varepsilon$ (ppm) | Keeling | | | Miller/Tans | | |
| | 100 | 500 | 1000 | 100 | 500 | 1000 |
| | **0.001(1)** | **0.000(1)** | **0.000(1)** | **0.001(1)** | **0.000(1)** | **0.000(1)** |
| 1 | 0.018(1) | 0.001(1) | 0.000(1) | 0.018(1) | 0.000(1) | 0.000(1) |
| | -0.296(1) | -0.032(1) | -0.017(1) | -0.165(1) | -0.016(1) | -0.008(1) |
| | **-0.011(1)** | **-0.002(1)** | **-0.001(1)** | **0.001(1)** | **0.003(1)** | **0.002(1)** |
| 5 | 0.427(1) | 0.019(1) | 0.006(1) | 0.413(1) | 0.012(1) | 0.002(1) |
| | -0.084(1) | -0.022(1) | -0.014(1) | 0.122(1) | -0.006(1) | -0.006(1) |
| | **-0.204(2)** | **-0.075(3)** | **-0.032(1)** | **-0.020(2)** | **-0.008(1)** | **0.001(1)** |
| 20 | 4.741(2) | 0.289(3) | 0.092(1) | 4.603(2) | 0.192(1) | 0.037(1) |
| | 2.386(2) | 0.127(3) | 0.037(1) | 3.399(2) | 0.135(1) | 0.022(1) |

**Table 3.** Monte Carlo (MC) and York estimates of the error in the isotopic source signature retrieved from an individual Keeling plot, in units of ‰. The mean goodness of fit $\chi^2_W$ over each 5000-fit ensemble is also shown.

| $\Delta c$ | $\varepsilon$ (ppm) | $\eta$ (‰) | MC | York | $\chi^2_W$ |
|---|---|---|---|---|---|
| 1 | 0.01 | 0.01 | 0.190 | 0.186 | 1.000 |
| | 0.01 | 0.15 | 2.80 | 2.79 | 1.000 |
| | 0.15 | 0.01 | 0.241 | 0.202 | 1.000 |
| | 0.2 | 0.3 | 6.84 | 4.60 | 0.999 |
| 10 | 0.01 | 0.01 | 0.0189 | 0.0189 | 1.000 |
| | 0.01 | 0.15 | 0.283 | 0.283 | 1.000 |
| | 0.15 | 0.01 | 0.0224 | 0.0221 | 1.000 |
| | 0.2 | 0.3 | 0.574 | 0.565 | 1.000 |
| 100 | 1 | 0.2 | 0.0425 | 0.0426 | 1.000 |
| | 20 | 0.2 | 0.153 | 0.147 | 0.986 |
| 1000 | 1 | 0.2 | 0.00797 | 0.00795 | 1.000 |
| | 20 | 0.2 | 0.0132 | 0.0131 | 0.996 |