# Peer review of "The long-solved problem of the best-fit straight line: Application to isotopic mixing lines"

_Biogeosciences, 2016_

## Referee Comment (RC1) · Anonymous Referee #1 · 10 Sep 2016

General Comments The manuscript by Wehr and Saleska re-introduces a non-linear iterative method to determine slope and intercepts of mixing line relationships. Isotopic mixing line relationships have been analyzed in previous studies (notably Zobitz et al. 2006 and Kayler et al. 2010). The current study expands on the previous two by introducing a "long-solved" method. Overall the paper is well written and readable.

A strength of the manuscript is that it very acutely emphasizes the disconnect between the geoscience and environmental science - and arguably the mathematical science - communities. Technical advances in one area don't seem to percolate over the the other area (as highlighted in the second paragraph of the introduction). Illuminating this tension between translatability across disciplines is a strength of the paper.

[Figure]

Specific Comments There are few weaknesses to the paper which could be addressed by revision.

First, there is a lack of recognition of the importance of OLS and other regression methods. OLS, GMR, ODR, as well as Maximum Likelihood Estimation are essentially a linear problem and are amenable to several different approaches in mathematics - OLS is a topic in a Calculus sequence. The York method, best I can tell, is a non-linear iterative method - which perhaps may contribute to its unfamiliarity across disciplines.

Second, I also think some more careful tracking of the timing of the key studies cited is important. Zobitz et al 2006 was written in response to previous studies by Pataki et al 2003. Kayler et al 2010 addressed minimizing bias for large CO2 ranges - and addressed some of the issues raised in the previous two studies. A consistent finding both in Zobitz et al 2006 and Kayler et al 2010 is that OLS is appropriate for sufficiently large CO2 ranges and is highly biased at low CO2 ranges. Given that, the authors of the current manuscript don't present a pressing need to move away from OLS in favor of a more complicated linear fitting method. What is the current state of the art in the measurement method? How imperative to determine mixing line parameters with samples of low CO2 ranges? Addressing some of the importance and need of this method will help increases its applicability, and the tradeoff for using a relatively more complex fitting routine than what is provided on all statistical software programs (R, SAS, etc).

Third, the results of this paper relied on subsampling of simulated data, which does limit the applicability of their results. I suggest the authors provide a case study of non-simulated data, comparing the two York method to OLS, GMR, ODR etc. Simulations are great for emphasizing the theoretical underpinnings of a method, however the addition of real measured data would enhance the applicability and impact of the simulation results.

Technical corrections (P = page, L = line)

P1 L21: "Much of it was outdated before it was written" is a very vague sentence. P1 L26: Point made that York's solution is unknown, but impact is not an indication of quality - I think it just got dwarfed, highlights the need for interdisciplinary collaboration.

P2 L9: Delete "but the debate is immaterial" This is a general sentence that is unprovable and opinion.

P3 L1: Please clarify if the Hirsch and Gilroy citation applies to all the quoted phrases in this sentence or only one (clarify)

P4 L27: Given the fact you need an initial guess slope, is the convergence of the method sensitive to the initial guess value, or does it converge globally?

P4 L24: "For CO2 ranges less than 50 ppm ..." This sentence reads very awkwardly and to follow the logic.. Please rephrase

P8 L22: Now I am confused. Does the York method give an exact solution (as in OLS, GMR) or is it a nonlinear iterative method as described on page 4?

---

## Referee Comment (RC2) · J. Miller (Referee) · 4 Oct 2016

Review of Wehr and Saleska, BGD, 2016, "The long-solved problem. . ."

Congratulations on writing a terrific paper: well researched, well written and very much needed. I have just a few questions and comments.

General comment:

How can we, or should we, consider variability in CO2 and $\delta$13C arising not from instrumental noise but from the environment? As pointed out in Miller and Tans, in some real-world situations, assignment of analytical uncertainties to CO2 and $\delta$13C may result in poor goodness of fit, i.e. a large value of reduced-chi square, suggesting that analytical

CO2 and δ13C uncertainties are too small. This is of course important because small CO2 and δ13C uncertainties will lead to too small slope and intercept uncertainties. Note that while not so common now, as analytical precision improves, instances where natural variability significantly exceeds instrumental precision will need to be dealt with more. In MT2003, we attempted to deal with this by starting with an initial estimate of the best fit line, although we used a GMR instead of fitexy (for speed, and because we only knew the analytical uncertainties). We then proceeded to scale the standard deviations of the x and y residuals to produce a reduced chi-square value of 1; finally fitexy was used to calculate the slope and intercept uncertainties. Nonetheless, a problem persists, which is that the slope of the best fit line depends on the initially assigned x and y uncertainties. I'm very interested to hear your ideas of how to address this. (Maybe I'm missing something obvious, like using an OLR regression as a starting point.)

Specific comments: P4 l8. Note for future reference that the Keeling plot equation is valid not just for a single source (or sink), but delta_s can be interpreted as a the flux-weighted source (sink) signature. Eq. 3. The derivation of this was not obvious. It's not critical to the argument, but since you have an appendix, can you add this? P7 l5. I am surprised by (and skeptical of) an instrument with 0.01 ppm and 0.01 per mil uncertainty. Can you provide a reference in the literature for this, especially since this is characterized as 'common instrumentation'? P7l8. Change 'latter' to 'last'. P7l34 and Table 2. I'm confused as to why CO2 ranges from 100 to 5000 are relevant and why CO2 uncertainties greater than 1 are relevant. I understand that soil chambers could give such high CO2 enhancements, but as seen from the table, uncertainties become very small. Perhaps you could add a column of 100 ppm in Table 1 and then summarize the rest of the Table2 results in the text. P8. L1. Why are the MT results a bit better at these high values? Or maybe better to say, why are the KP biases occasionally significant? P8l29. Factor of 2 seems a bit too generous. The biggest offset from Monte Carlo I see is 0.67. P8l35. What are the 'adjusted data points'? P9.l7 Isn't G simply reduced chi-square? If so, why introduce a new term for this?

John Miller

---

## Author Response (AR1)

Dear Dr. Jens-Arne Subke,

We are pleased to submit our revised manuscript, following your and the referees' suggestions, along with a marked up copy in which all changes since the original submission are detailed in red. Below you will also find our point-by-point responses to your comments and to the referees' comments (note that you have already read our responses to the referees' comments; we include them again here only because the instructions for this stage say we should). We have not included a separate list of all changes because it would be redundant with (and surely less useful than) the marked up manuscript.

We thank you for your efforts on our manuscript. Please let us know if you would like anything else from us.

Best Regards,

Rick Wehr (also on behalf of Scott Saleska)

**Response to the Associate Editor's Decision on bg-2016-315**

**Response of authors Wehr and Saleska (hereafter "WS) to Associate Editor Jens-Arne Subke (hereafter "JS"):**

*JS: Many thanks for your replies to the two referee reports. Both referees have made an excellent job of appraising your manuscript critically, and together with your response to their individual points raised, I think it has made this publication even stronger. I'm sure that in its final form, it will become an important reference for isotope biogeochemists.*

WS: Thank you. We agree that the referee's and your comments have improved the manuscript, and we hope that it will be well received by readers.

*JS: I'd like you to implement the suggested revision points, including a brief discussion of the issue of instrument uncertainty vs. variability caused by spatial/ecosystem heterogeneity, raised by referee 2 (John Miller).*

WS: We have implemented all the suggested revision points in our revised manuscript, including the addition of a new section discussing natural variability (Section 2.2).

*JS: The new figure illustrating a 'real data' example of night-time Keeling plots is also very useful. A detailed figure caption outlining the differences in data sets would obviously be necessary alongside it.*

We have included the new figure in our revised manuscript, with a caption and a corresponding discussion in the text (Section 4.2).

**Response to Referee Comments on bg-2016-315 (already seen by the Associate Editor)**

**Author Response to Anonymous Referee #1 [hereafter "AR1"]**

*AR1: The manuscript by Wehr and Saleska re-introduces a non-linear iterative method to determine slope and intercepts of mixing line relationships. Isotopic mixing line relationships have been analyzed in previous studies (notably Zobitz et al. 2006 and Kayler et al. 2010). The current study expands on the previous two by introducing a "long-solved" method. Overall the paper is well written and readable. A strength of the manuscript is that it very acutely emphasizes the disconnect between the geoscience and environmental science - and arguably the mathematical science - communities. Technical advances in one area don't seem to percolate over the the other area (as highlighted in the second paragraph of the introduction). Illuminating this tension between translatability across disciplines is a strength of the paper.*

We thank the referee for these supportive comments, and for taking the time to review our manuscript.

*AR1: There are few weaknesses to the paper which could be addressed by revision.*

We appreciate the referee's suggestions and will incorporate them into our revised manuscript as described below.

*AR1: First, there is a lack of recognition of the importance of OLS and other regression methods. OLS, GMR, ODR, as well as Maximum Likelihood Estimation are essentially a linear problem and are amenable to several different approaches in mathematics - OLS is a topic in a Calculus sequence. The York method, best I can tell, is a non-linear iterative method - which perhaps may contribute to its unfamiliarity across disciplines.*

We agree on "the importance of OLS and other regression methods", and had intended that the manuscript acknowledge their wide use while making clear that their overuse (in cases where their assumptions do not apply) is an explicit motivation for the paper. Regarding the contrast between methods, yes, the general LSE solution for the slope and intercept cannot be written analytically, and so York's method finds it numerically, by iteration. The OLS, GMR, and ODR slopes and intercepts are special cases of the general LSE solution that can be solved for analytically. We can mention this distinction in our revised manuscript.

*AR1: Second, I also think some more careful tracking of the timing of the key studies cited is important. Zobitz et al 2006 was written in response to previous studies by Pataki et al 2003. Kayler et al 2010 addressed minimizing bias for large CO2 ranges -*

*and addressed some of the issues raised in the previous two studies. A consistent finding both in Zobitz et al 2006 and Kayler et al 2010 is that OLS is appropriate for sufficiently large CO2 ranges and is highly biased at low CO2 ranges. Given that, the authors of the current manuscript don't present a pressing need to move away from OLS in favor of a more complicated linear fitting method. What is the current state of the art in the measurement method? How imperative to determine mixing line parameters with samples of low CO2 ranges? Addressing some of the importance and need of this method will help increases its applicability, and the tradeoff for using a relatively more complex fitting routine than what is provided on all statistical software programs (R, SAS, etc).*

We like the referee's suggestion to add mention of the timing/context/motivation of the cited studies, and we will do so in our revised manuscript. Zobitz et al 2006 actually reported that OLS was negligibly biased for all measurement conditions (but the random error increased at low CO2 ranges). On the other hand, Kayler et al 2010 reported that the OLS Keeling plot intercept was negligibly biased (i.e. by less than about 0.1 permil) for CO2 ranges above roughly 50 ppm but non-negligibly biased for CO2 ranges below roughly 10 ppm. The particular numbers depend on the measurement precisions (as illustrated by our tables); however, in Kayler et al 2010, various precision scenarios seem to have been amalgamated in their Table 1, so that for a given set of measurement precisions, it is not clear whether non-negligible OLS bias emerges at a CO2 range of 100 ppm or 10 ppm. That is an important issue because Keeling plots with CO2 ranges below 50 ppm are not unusual — indeed the real data example that we will include in our revised manuscript (see below) involves hundreds of Keeling plots made from nighttime air sampling in a forest, and most of those Keeling plots have a CO2 range below 50 ppm (for quite a few, the range is even below 25 ppm). So prior to the present manuscript, it was unclear whether it was acceptable to use OLS under some fairly common conditions. Moreover, the primary conclusion of Kayler et al 2010 was actually that GMR is better than OLS for very high CO2 ranges such as might be encountered in soil measurements ("The combination of geometric mean regression and the Miller–Tans mixing model provided the most accurate and precise estimate of d13CR when the range of CO2 is >1,000 umol mol-1."). The main goal of our paper is to inform readers that there is a single general fit method (York's) that is best in all measurement scenarios, so that they do not need to make a choice among imperfect methods or to switch between those methods depending on the conditions. We will add a paragraph to the introduction clarifying the above points, thereby adding motivation for the use of York's method.

Having said all that, we recognize that OLS is more accessible and familiar to researchers, and we do not argue that researchers must abandon OLS. Indeed, a second goal of our paper is to accurately detail the range of conditions in which OLS can safely be used to approximate the York solution (hence the tables with biases under various conditions). It turns out that under most practical conditions, OLS is an acceptable approximation to the York solution, while under some conditions, it is not. We will clarify this point in the Results and discussion section. When in doubt,

researchers will now have a general method that they can use without risking introducing bias.

More broadly, the isotopic mixing line is being used here as a practical example to illustrate the York method, which might prove even more valuable to other applications, quite apart from our own line of research. We would like to add a paragraph briefly presenting a separate potential application of York's solution, which is quite common across disciplines: the comparison of measurements of the same quantity by two different instruments that differ in their precisions (such as when replacing an old instrument with a new one).

*AR1: Third, the results of this paper relied on subsampling of simulated data, which does limit the applicability of their results. I suggest the authors provide a case study of non-simulated data, comparing the two York method to OLS, GMR, ODR etc. Simulations are great for emphasizing the theoretical underpinnings of a method, however the addition of real measured data would enhance the applicability and impact of the simulation results.*

As mentioned above, we have a real data set that we will add to the manuscript: precise (0.05 permil and 0.02 ppm) nighttime air sampling on the tower at the Harvard Forest Environmental Measurements Site. Given that our simulated data results show the York method to be unbiased, we can suppose that the bias in GMR (or OLS) is given by the difference between the GMR (or OLS) intercept and the York intercept. The 429 nights in our dataset (with one Keeling plot per night) allow us to clearly see the dependence of the biases on the CO2 range — and by adding additional noise to the real data, we also get an indication of the dependence of the bias on the instrument precision. We find that OLS is a very good approximation to the York solution for our particular instrument precision, while GMR is a poor one. With noise levels typical of lower-cost instruments, we find that even OLS becomes a poor approximation for $CO_2$ ranges less than 10 ppm, in agreement with our simulated data results. These results using real data are shown in the figure attached to this response.

*AR1: P1 L21: "Much of it was outdated before it was written" is a very vague sentence.*

Taken in isolation, it is vague, but this sentence is the deliberately suspenseful setup for the two sentences that follow, which make the point that the solution sought by the literature in question was already known in 1969. In context, we feel that the sentence is not vague, and that it adds to the readability of the manuscript.

*P1 L26: Point made that York's solution is unknown, but impact is not an indication of quality - I think it just got dwarfed, highlights the need for interdisciplinary collaboration.*

We are glad that the referee agrees with us on this motivational point.

*AR1: P2 L9: Delete "but the debate is immaterial" This is a general sentence that is unprovable and opinion.*

We meant to convey the idea that when a convenient, general solution is available, it is not necessary to choose among special-case solutions. But perhaps we do not need to make that point here, and so we will delete this phrase.

*AR1: P3 L1: Please clarify if the Hirsch and Gilroy citation applies to all the quoted phrases in this sentence or only one (clarify)*

We will clarify by writing: "Adding confusion to the literature is the fact that OLS, ODR, and GMR are each known by other names (York, 1966; Hirsch and Gilroy, 1984): OLS is called….".

*AR1: P4 L27: Given the fact you need an initial guess slope, is the convergence of the method sensitive to the initial guess value, or does it converge globally?*

The method will require more iterations to converge to the desired accuracy level when the initial guess is poorer, but the value of the initial guess has little impact on the convergence speed unless the guess is extreme. As stated in the manuscript, "This guess can be very rough and still sufficient… If desired, a good initial guess slope can be obtained from an OLS fit".

*AR1: P7 L24: "For CO2 ranges less than 50 ppm . . ." This sentence reads very awkwardly and to follow the logic.. Please rephrase*

We can rephrase the sentence thusly: "For CO2 ranges less than 50 ppm, the GMR bias is non-negligible (> 0.1 ‰) unless the measurement uncertainty in $\delta$ is extraordinarily low (≤ 0.01 ‰)."

*AR1: P8 L22: Now I am confused. Does the York method give an exact solution (as in OLS, GMR) or is it a nonlinear iterative method as described on page 4?*

We see that the meaning of 'exact' was unclear. York's general LSE solution is exact in that no approximation is involved in its derivation, unlike the preceding approach of Deming (1943), which dropped terms from a Taylor expansion. As mentioned above, York's solution is also a numerical, iterative solution, in contrast to the special-case LSE solutions (OLS, ODR, and GMR), which can be written analytically (i.e. in closed form). In practice, there is no difference in exactitude between the numerical approach of York and the analytical approach of OLS, ODR, and GMR; all are limited only by machine precision. This will be clarified in the revision.

**Author Response to John Miller (Referee) [hereafter "JM"]**

*JM: Congratulations on writing a terrific paper: well researched, well written and very much*
*needed. I have just a few questions and comments.*

We thank the referee for his compliments and for taking the time to review our paper.

*JM: How can we, or should we, consider variability in CO2 and δ13C arising not from instrumental noise but from the environment? As pointed out in Miller and Tans, in some real- world situations, assignment of analytical uncertainties to CO2 and δ13C may result in poor goodness of fit, i.e. a large value of reduced-chi square, suggesting that analytical CO2 and δ13C uncertainties are too small. This is of course important because small CO2 and δ13C uncertainties will lead to too small slope and intercept uncertainties. Note that while not so common now, as analytical precision improves, instances where natural variability significantly exceeds instrumental precision will need to be dealt with more. In MT2003, we attempted to deal with this by starting with an initial estimate of the best fit line, although we used a GMR instead of fitexy (for speed, and because we only knew the analytical uncertainties). We then proceeded to scale the standard deviations of the x and y residuals to produce a reduced chi-square value of 1; finally fitexy was used to calculate the slope and intercept uncertainties. Nonetheless, a problem persists, which is that the slope of the best fit line depends on the initially assigned x and y uncertainties. I'm very interested to hear your ideas of how to address this. (Maybe I'm missing something obvious, like using an OLR regression as a starting point.)*

As you know, York's method deals with the situation in which there is a linear relationship between the true values of X and Y (i.e. of CO2 and d13C) but those values are measured with error. Natural variability may sometimes be describable as measurement error; that is, as a stochastic process that intervenes between the quantity of interest and the measurement of that quantity. For example, the eddy covariance method uses a single-point measurement to estimate the gas flux through a large 2D plane overlying an ecosystem, and most of the noise in the estimation comes not from the instrumental measurement uncertainty but from the natural (turbulence-driven) variability in the flux past that single point relative to the flux through the whole 2D plane. If the natural variability in X and the natural variability in Y are describable as measurement error and can be characterized independently (along with any correlation between them), then York's method can be applied and is likely to be very useful.

On the other hand, it is often the case that the natural variability is not well characterized, or that it is not well described as additional measurement error. In this case, we argue that one cannot proceed to determine the best-fit line, or even to define what "best-fit" means. In general, one can view natural variability as variation in the true X-Y relationship due to the influence of other factors that are not controlled for. So a Keeling plot with natural variability is like many true mixing lines all superimposed on the same plot (one line for each set of influencing factors).

It is therefore pertinent to consider which true line one is looking for. To define that line is, in effect, to characterize the natural variability in X and Y.

If one is interested not in the X-Y relationship *per se* (i.e. not in an intercept or slope), but simply in predicting the most likely value of Y given X for the particular data that were sampled and put on the plot, then OLS is the proper fit to use. If differences among the various fit methods are not large enough to matter to the scientific question being posed, then OLS is again a sensible choice, owing to its simplicity.

If it would be helpful for context, we would be happy to include a brief discussion of the above points in the manuscript.

Regarding the approach you mention from MT2003, it seems that the slope you end up with must depend on your arbitrary choice of how to apportion the variability into X and Y. That seems to be what you are saying when you say that "the slope of the best fit line depends on the initially assigned x and y uncertainties". Any meaningful approach will require independent information on how to apportion the natural variability between X and Y.

*JM: P4 l8. Note for future reference that the Keeling plot equation is valid not just for a single source (or sink), but delta_s can be interpreted as a the flux- weighted source (sink) signature.*

That is a good point. We will remove the word "single" from this sentence.

*JM: Eq. 3. The derivation of this was not obvious. It's not critical to the argument, but since you have an appendix, can you add this?*

Yes, we will be happy to add the derivation of Eq. 3. It consists merely of writing the definition of the correlation coefficient in terms of the covariance, and then the definition of the covariance in terms of means and expectations, and so it is probably short enough to be included in the main text.

*JM: P7 l5. I am surprised by (and skeptical of) an instrument with 0.01 ppm and 0.01 per mil uncertainty. Can you provide a reference in the literature for this, especially since this is characterized as 'common instrumentation'?*

The 0.01 ppm and 0.01 permil uncertainties are attributed to "the best existing laser spectrometer" rather than common instrumentation, but we recognize that this point is confusing because of the placement of the phrase "including some corresponding to common instrumentation" (which was meant to apply to the whole list). We will move that phrase to avoid the confusion. The spectrometer referred to has a precision of 0.016 ppm and 0.02 permil under optimal conditions, and we will add its citation (Wehr et al, 2013, Agricultural and Forest Meteorology, 181,69–84). We will also clarify that 0.01 and 0.01 are slightly better than said

spectrometer (we chose to round down to 0.01 because our aim was to bracket the range of conditions that researchers are likely to encounter, and spectrometer precision is likely to continue to improve).

*JM: P7l8. Change 'latter' to 'last'.*

Ok, we will make this change.

*JM: P7l34 and Table 2. I'm confused as to why CO2 ranges from 100 to 5000 are relevant and why CO2 uncertainties greater than 1 are relevant. I understand that soil chambers could give such high CO2 enhancements, but as seen from the table, uncertainties become very small. Perhaps you could add a column of 100 ppm in Table 1 and then summarize the rest of the Table2 results in the text.*

We included those ranges and uncertainties for comparison to Kayler et al 2010, where it is argued that high CO2 values are often accompanied by high uncertainties. We agree that the 5000 ppm column is unnecessary, but for the highest CO2 uncertainty (20 ppm), there are non-negligible biases even for a CO2 range of 1000 ppm. We can condense the table substantially by eliminating the 5000 ppm column and then putting the Keeling and Miller/Tans results side by side rather than on top of one another.

*JM: P8. L1. Why are the MT results a bit better at these high values? Or maybe better to say, why are the KP biases occasionally significant?*

This question is the subject of the following paragraph in the manuscript (beginning on P8 L4).

*JM: P8l29. Factor of 2 seems a bit too generous. The biggest offset from Monte Carlo I see is 0.67.*

We can say that "the agreement is nonetheless within 33%."

*JM: P8l35. What are the 'adjusted data points'?*

The adjusted data points are the fit method's estimation of the 'true' data points that were measured with error in order to produce the measured data points. We will add this explanation to the manuscript.

*JM: P9.l7 Isn't G simply reduced chi-square? If so, why introduce a new term for this?*

G is the weighted reduced chi-square. In our revised manuscript, we will state that this is the goodness of fit metric being used, and we will use the variable chi-square in place of G.

[revised manuscript text omitted]

Richard Wehr 11/14/2016 4:56 PM
**Comment [2]:** This discussion on historical context added at referee's request

Richard Wehr 11/1/2016 9:27 AM

Richard Wehr 11/1/2016 9:27 AM

Richard Wehr 11/1/2016 10:07 AM

[revised manuscript text omitted]

Richard Wehr 11/1/2016 9:45 PM

Richard Wehr 11/1/2016 9:45 PM

Richard Wehr 11/1/2016 9:46 PM

Richard Wehr 11/1/2016 3:25 PM

Richard Wehr 11/1/2016 3:28 PM

Richard Wehr 11/2/2016 10:17 AM

Richard Wehr 11/1/2016 8:57 AM

The York equations, on the other hand, produce unbiased Keeling and Miller/Tans fit lines for all conditions in the table. Because the emergence of high-frequency isotopic measurements is starting to raise the issue, we show in detail how some OLS- and York-retrieved isotopic source signatures compare at the lowest $\Delta c$ in Figure 2, where the error bars represent $\pm 2\sigma$, i.e. twice the standard error in the mean of 5000 fits.

Isotopic source signatures retrieved from our simulated Keeling and Miller/Tans plots for $\Delta c \geq 100$ ppm are reported in Table 2. Again, the York method is by far the best, although the York fit lines do exhibit small but detectible biases for some sets of conditions here. The York Miller/Tans fits are biased by at most -0.020 ‰, while bias in the York Keeling fits is worse, reaching -0.204 ‰ when $\varepsilon = 20$ ppm (an exceptionally high value) and $\Delta c = 100$ ppm. This bias is still an order of magnitude less than the bias from any of the other methods under those conditions. The York Miller/Tans bias is due to the approximations made in Eqs. (1) and (5). We have confirmed this by comparing simulations with and without the approximations (a luxury not available with real data). The Keeling bias is due partly to the approximation in Eq. (2), but mostly to the non-normal error distribution in $1/\hat{c}$ (see Section 3.1): a distribution that is asymmetric, with a non-zero mean. In Table 1, where $\varepsilon$ (maximum 0.15 ppm) is always a small fraction of $c$ (380 ppm), the skew of the error in $1/\hat{c}$ is small and its effect on the fit negligible; however, in Table 2, where $\varepsilon/c$ can be as large as 5 %, the skew becomes relatively large (the uncertainty on one side of a point is about 10 % larger than on the other side) and its effect on the fit becomes detectible in our simulations. The bias induced in the fit by the non-normal error distribution should increase as $\varepsilon$ increases and as $\Delta c$ decreases, which are the trends we observe. The preceding explanation is confirmed by the fact that when we alter our simulations by adding normally distributed measurement error directly to $1/c$ rather than to $c$ (and giving the correct information to the York fitting algorithm), we find that the York Keeling fits are completely unbiased (results not tabulated). Luckily, the York fit biases we report in Table 2 are very small considering the measurement uncertainties necessary to induce them, and are unlikely to be the limiting source of error in any experiment.

We also tested *fitexy* using our Monte Carlo simulations. As expected, the *fitexy* results were always identical to the York results when fitting to Keeling plots but were in error when fitting to Miller/Tans plots, because the latter plots involved correlation between the $x$ and $y$ errors. For example, with the fairly large measurement uncertainties $\varepsilon = 0.2$ ppm and $\eta = 0.3$ ‰, the *fitexy* Miller/Tans slopes were biased by -4.259 ‰ for $\Delta c = 1$ ppm and -0.027 ‰ for $\Delta c = 10$ ppm.

**4.2 Comparison of fit biases using real measurements**

The intercepts from OLS, GMR, and York fits to our measured Keeling plots are compared in Figure 3, as a function of the $CO_2$ range. Given that our Monte Carlo simulations show the York fit to be unbiased, we can use the difference between the OLS (or GMR) and York intercepts as a proxy for the bias in OLS (or GMR). In agreement with our Monte Carlo results, Figure 3 shows that for the original measurement uncertainties of 0.05 ppm and 0.02 ‰, GMR is negligibly biased (that is, by less than 0.1 ‰) only for $CO_2$ ranges above about 25 ppm, while OLS is negligibly biased for all $CO_2$ ranges. Also in agreement with our Monte Carlo results, the figure shows that if the measurement uncertainties are increased to the more common values of 0.2 ppm and 0.3 ‰, then GMR is never negligibly biased, while OLS is negligibly biased only for $CO_2$ ranges above 10 ppm. Note that the scatter in Figure 3 is due to the fact that unlike our simulated data points, our real measured data points were not evenly distributed throughout the $CO_2$ range; in some of the measured Keeling plots, almost all of the points were clustered in a small portion of the range, leading to a higher bias.

Richard Wehr 11/2/2016 10:18 AM

Richard Wehr 11/2/2016 10:19 AM

Richard Wehr 11/2/2016 10:23 AM

Richard Wehr 11/2/2016 10:21 AM

Richard Wehr 11/14/2016 5:06 PM
**Comment [3]:** New section related to the measured data figure

[revised manuscript text omitted]

Richard Wehr 11/14/2016 4:55 PM
**Comment [4]:** Figure now uses real measured data

Richard Wehr 11/1/2016 3:08 PM

Richard Wehr 11/1/2016 3:31 PM

Richard Wehr 11/1/2016 3:31 PM

Richard Wehr 11/1/2016 3:08 PM

Richard Wehr 11/1/2016 3:08 PM

Richard Wehr 11/2/2016 10:26 PM

Richard Wehr 11/2/2016 10:25 PM

Richard Wehr 11/1/2016 3:32 PM

Richard Wehr 11/1/2016 3:08 PM

[Figure]

Figure 2: **Mean isotopic signatures retrieved from ensembles of 5000 simulated Keeling plots (each containing 5000 points) using the York and OLS methods, for $CO_2$ ranges from 1 to 10 ppm and a true isotopic signature of -25 ‰. The standard deviations of the random noise added to the simulated measurements of $c$ and $\delta$ are inset. Error bars show two standard errors. Note the differing scales for the ordinate.**

[Figure]

Figure 3: **Difference between the fit intercept obtained by OLS or GMR and that obtained by York's equations, for 429 real measured Keeling plots with measurement uncertainties of 0.05 ppm and 0.02 ‰ and with $CO_2$ mixing ratio ranges between 2 and 335 ppm. Also shown is the difference obtained for Keeling plots in which random computer-generated noise of 0.2 ppm and 0.3 ‰ was added to the original data. The grey region represents biases that are negligible for most practical purposes (< 0.1 ‰).**

Richard Wehr 11/14/2016 4:55 PM

**Comment [5]:** New figure

[Figure]

Figure 4: Two lines, each defined by measurements of the true points (2,2) and (6,6) that are in error by +1 or -1 in the ordinate only.

Richard Wehr 11/1/2016 12:06 PM

[Figure]

Figure 5: Distributions of the $y$-intercepts retrieved from Keeling plots with a true intercept of -25 ‰ (indicated by the vertical line). The black curve was obtained from $10^6$ two-point plots. The red curves were obtained using the York equations on $10^6$ five-point or twenty-point plots. The blue curves were obtained using OLS on $10^6$ five-point or twenty-point plots. The heights of the curves have been adjusted independently for presentation and convey no meaning.

Richard Wehr 11/1/2016 12:06 PM

**Table 1. Bias in retrieved isotopic source signatures from Keeling line fits for $\Delta c \leq 50$ ppm, expressed as differences from the true value of -25, in units of ‰. Each cell contains, from top to bottom: the York, the OLS, and the GMR result. 1-$\sigma$ uncertainties in the final digit(s) are given in parentheses, based on the standard deviation of the ensemble of 5000 fits.**

| $\varepsilon$ (ppm) | $\eta$ (‰) | CO$_2$ Range (ppm) | | | |
|---|---|---|---|---|---|
| | | 1 | 5 | 10 | 50 |
| 0.01 | 0.01 | **0.005(3)** | **0.000(1)** | **0.000(1)** | **0.000(0)** |
| | | 0.024(3) | 0.001(1) | 0.000(1) | 0.000(0) |
| | | -4.700(3) | -0.214(1) | -0.054(1) | 0.002(0) |
| 0.01 | 0.15 | **-0.066(40)** | **-0.006(8)** | **0.007(4)** | **0.000(1)** |
| | | -0.047(40) | -0.005(8) | 0.008(4) | 0.000(1) |
| | | -181.95(3) | -26.592(6) | -9.400(4) | -0.477(1) |
| 0.05 | 0.05 | **0.010(14)** | **0.001(3)** | **0.004(2)** | **0.000(1)** |
| | | 0.475(13) | 0.020(3) | 0.009(2) | 0.000(1) |
| | | -50.840(10) | -4.781(3) | -1.343(2) | -0.065(1) |
| 0.15 | 0.01 | **0.002(4)** | **0.000(1)** | **0.000(1)** | **0.000(0)** |
| | | 3.398(3) | 0.171(1) | 0.043(1) | 0.002(0) |
| | | -2.385(3) | -0.128(1) | -0.032(1) | -0.001(0) |
| 0.15 | 0.15 | **-0.072(44)** | **-0.005(9)** | **0.001(4)** | **0.000(1)** |
| | | 3.339(35) | 0.165(8) | 0.043(4) | 0.002(1) |
| | | -160.00(3) | -26.673(6) | -9.657(4) | -0.575(1) |
| 0.2 | 0.3 | **0.108(97)** | **-0.028(16)** | **-0.006(8)** | **0.002(2)** |
| | | 5.251(65) | 0.272(16) | 0.070(8) | 0.005(2) |
| | | -303.09(89) | -64.388(12) | -26.100(6) | -2.185(2) |

**Table 2.** Bias in retrieved isotopic source signatures from Keeling and Miller/Tans line fits for $\Delta c \geq 100$ ppm and $\eta = 0.2$ ‰, expressed as differences from the true value of –25, in units of ‰. Each cell contains, from top to bottom: the York, the OLS, and the GMR result. 1-$\sigma$ uncertainties in the final digit are given in parentheses, based on the standard deviation of the ensemble of 5000 fits.

Richard Wehr 11/2/2016 10:06 AM
**Comment [6]:** Table has been rearranged, dropping out the 5000 ppm column.

| $\varepsilon$ (ppm) | CO$_2$ Range (ppm) | | | | | |
| --- | --- | --- | --- | --- | --- | --- |
| | Keeling | | | Miller/Tans | | |
| | 100 | 500 | 1000 | 100 | 500 | 1000 |
| 1 | **0.001(1)** | **0.000(1)** | **0.000(1)** | **0.001(1)** | **0.000(1)** | **0.000(1)** |
| | 0.018(1) | 0.001(1) | 0.000(1) | 0.018(1) | 0.000(1) | 0.000(1) |
| | -0.296(1) | -0.032(1) | -0.017(1) | -0.165(1) | -0.016(1) | -0.008(1) |
| 5 | **-0.011(1)** | **-0.002(1)** | **-0.001(1)** | **0.001(1)** | **0.003(1)** | **0.002(1)** |
| | 0.427(1) | 0.019(1) | 0.006(1) | 0.413(1) | 0.012(1) | 0.002(1) |
| | -0.084(1) | -0.022(1) | -0.014(1) | 0.122(1) | -0.006(1) | -0.006(1) |
| 20 | **-0.204(2)** | **-0.075(3)** | **-0.032(1)** | **-0.020(2)** | **-0.008(1)** | **0.001(1)** |
| | 4.741(2) | 0.289(3) | 0.092(1) | 4.603(2) | 0.192(1) | 0.037(1) |
| | 2.386(2) | 0.127(3) | 0.037(1) | 3.399(2) | 0.135(1) | 0.022(1) |

**Table 3. Monte Carlo (MC) and York estimates of the error in the isotopic source signature retrieved from an individual Keeling plot, in units of ‰. The mean goodness of fit $\chi^2_W$ over each 5000-fit ensemble is also shown.**

| $\Delta c$ | $\varepsilon$ (ppm) | $\eta$ (‰) | MC | York | $\chi^2_W$ |
|---|---|---|---|---|---|
| 1 | 0.01 | 0.01 | 0.190 | 0.186 | 1.000 |
| | 0.01 | 0.15 | 2.80 | 2.79 | 1.000 |
| | 0.15 | 0.01 | 0.241 | 0.202 | 1.000 |
| | 0.2 | 0.3 | 6.84 | 4.60 | 0.999 |
| 10 | 0.01 | 0.01 | 0.0189 | 0.0189 | 1.000 |
| | 0.01 | 0.15 | 0.283 | 0.283 | 1.000 |
| | 0.15 | 0.01 | 0.0224 | 0.0221 | 1.000 |
| | 0.2 | 0.3 | 0.574 | 0.565 | 1.000 |
| 100 | 1 | 0.2 | 0.0425 | 0.0426 | 1.000 |
| | 20 | 0.2 | 0.153 | 0.147 | 0.986 |
| 1000 | 1 | 0.2 | 0.00797 | 0.00795 | 1.000 |
| | 20 | 0.2 | 0.0132 | 0.0131 | 0.996 |